# Variability of Mitochondrial DNA Heteroplasmy: Association with Asymptomatic Carotid Atherosclerosis

**DOI:** 10.3390/biomedicines12081868

**Published:** 2024-08-15

**Authors:** Margarita A. Sazonova, Tatiana V. Kirichenko, Anastasia I. Ryzhkova, Marina D. Sazonova, Natalya A. Doroschuk, Andrey V. Omelchenko, Nikita G. Nikiforov, Yulia I. Ragino, Anton Yu. Postnov

**Affiliations:** 1Laboratory of Angiopathology, Institute of General Pathology and Pathophysiology, 8 Baltiiskaya Street, Moscow 125315, Russia; ryzhkovaai@gmail.com (A.I.R.); marinasazon1990@gmail.com (M.D.S.); natador28@mail.ru (N.A.D.); omi@bk.ru (A.V.O.); 2Laboratory of Medical Genetics, Institute of Experimental Cardiology, Chazov National Medical Research Center of Cardiology, 15a, 3rd Cherepkovskaya Str., Moscow 121552, Russia; anton-5@mail.ru; 3Laboratory of Cellular and Molecular Pathology of Cardiovascular System, Federal State Budgetary Scientific Institution, Petrovsky National Research Centre of Surgery (FSBSI “Petrovsky NRCS”), Moscow 117418, Russia; t-gorchakova@mail.ru (T.V.K.); nikiforov.mipt@googlemail.com (N.G.N.); 4Research Institute of Internal and Preventive Medicine—Branch of the Institute of Cytology and Genetics, Siberian Branch of Russian Academy of Sciences, Novosibirsk 630089, Russia; ragino@mail.ru

**Keywords:** atherosclerosis, atherosclerotic plaque, mitochondrion, gene, mutation, pyrosequencing, heteroplasmy level, total mutational burden, cardiovascular risk

## Abstract

**Background and Objectives:** Atherosclerosis is one of the main reasons for cardiovascular disease development. This study aimed to analyze the association of mtDNA mutations and atherosclerotic plaques in carotid arteries of patients with atherosclerosis and conditionally healthy study participants from the Novosibirsk region. **Methods:** PCR fragments of DNA containing the regions of 10 investigated mtDNA mutations were pyrosequenced. The heteroplasmy levels of mtDNA mutations were analyzed using a quantitative method based on pyrosequencing technology developed by M. A. Sazonova and colleagues. **Results:** In the analysis of samples of patients with atherosclerotic plaques of the carotid arteries and conditionally healthy study participants from the Novosibirsk region, four proatherogenic mutations in the mitochondrial genome (m.5178C>A, m.652delG, m.12315G>A and m.3256C>T) and three antiatherogenic mutations in mtDNA (m.13513G>A, m.652insG, and m.14846G>A) were detected. A west–east gradient was found in the distribution of the mtDNA mutations m.5178C>A, m.3256C>T, m.652insG, and m.13513G>A. **Conclusions:** Therefore, four proatherogenic mutations in the mitochondrial genome (m.5178C>A, m.652delG, m.12315G>A, and m.3256C>T) and three antiatherogenic mutations in mtDNA (m.13513G>A, m.652insG, and m.14846G>A) were detected in patients with atherosclerotic plaques in their carotid arteries from the Novosibirsk region.

## 1. Introduction

Cardiovascular disease (CVD) is one of the leading causes of premature mortality in industrially developed countries [1,2,3,4,5]. In numerous cases, CVD is caused by atherosclerosis. Currently, there are many methods for the non-invasive and invasive diagnosis of atherosclerosis such as MRI, B-mode ultrasound, IVUS, and invasive and CT angiography [6,7,8,9,10]. However, these measures are still insufficient for the timely diagnosis of early atherosclerotic lesions and for the prevention of atherosclerosis development. In addition, the choice of therapeutic agents for anti-atherosclerotic therapy is quite limited [11,12,13,14,15]. Furthering the search for novel markers and factors to develop new strategies in the prevention and treatment of atherosclerosis and atherosclerosis-based diseases is extremely relevant. Traditional cardiovascular risk factors (obesity, hypercholesterolemia, high blood pressure, smoking, etc.) do not fully explain the variability in the clinical manifestations of atherosclerosis [16,17,18,19,20]. Therefore, the identification of genetic markers for the diagnosis of predisposition to CVD and atherosclerosis is a promising area of scientific research.

Mitochondrial genome mutations are believed to be perspective markers for the prognosis of atherosclerosis and CVD development [21,22,23,24,25]. mtDNA is capable of accumulating mutations throughout the life of an individual because of its weak reparative system. This is facilitated by the excessive generation of reactive oxygen species and the effect of free radicals on the mitochondrial genome [26,27,28,29,30]. Since there are several mitochondria in a cell, it may have several copies of the mitochondrial genome, so the distribution of mutant copies of mtDNA throughout the body, organ, or tissue is uneven [31,32,33,34,35]. It is well known that mitochondrial genome mutations are associated with various pathologies, including ischemic heart disease, various types of cardiomyopathies, neurodegenerative diseases, diabetes mellitus, and oncological diseases [36,37,38,39,40]. However, studies presenting data on the association of mtDNA mutations with atherosclerosis are rare. Until now, mutations of the mitochondrial genome have been insufficiently evaluated as diagnostic markers of CVD and atherosclerosis. The ten mtDNA mutations analyzed in this study were selected because they have been the subject of previous studies in the Moscow region. The aim of the present study is the detection of regional mtDNA markers associated with atherosclerotic plaques (ASPs) in the carotid arteries of patients from the Novosibirsk region. The present study is replicative, and its second aim is to find the direction of the gradient of distribution of the analyzed mutations.

## 2. Materials and Methods

### 2.1. Study Participants

The study was performed in accordance with the principles outlined in the Declaration of Helsinki of 1975 and its revised version of 2013. The study protocol was approved by the Institute for Atherosclerosis Research Committee on Human Research, Moscow, Russia as protocol No. 078-15 on 8 September 2015. The participants of the study were recruited from the Research Institute of Internal and Preventive Medicine, Novosibirsk, Russia. Inclusion criteria were an age of 55–79 years and the absence of clinical manifestations of CVD. In addition, each patient included in the study signed an informed consent form. Exclusion criteria were myocardial infarction, angina pectoris, intermittent claudication, transient ischemic attacks, arterial revascularization, and stroke.

We created two samples of the study participants:

(1) Patients with atherosclerotic plaques in their carotid arteries;

(2) Conditionally healthy study participants.

The major estimated parameter was the level of heteroplasmy of mitochondrial genome mutations previously detected as pro- or anti-atherogenic [41]. The following traditional cardiovascular risk factors were also assessed in each participant to characterize the study population: arterial blood pressure, body mass index, presence of diabetes mellitus, status of smoking, blood lipid profile, and thickness of the intima–medial layer of the carotid arteries.

### 2.2. Ultrasonography of Carotid Arteries

To evaluate the state of the carotid artery walls, high-resolution B-mode ultrasonography was performed using the SonoScape SSI-1000 ultrasound scanner with a linear array vascular probe 7.5 MHz (SonoScape Medical Corp., Shanghai, China). Participants with ASPs in any visualized segment of the carotid arteries were included in the first sample; participants without any ultrasound signs of atherosclerosis lesions in their carotid arteries were included in the second sample. To assess the presence of ASPs, the left and right common carotid arteries, the carotid sinus area, as well as the external and internal carotid arteries were examined in three fixed projections—anterior, lateral, and posterior. An increase in the thickness of the intima–medial layer of >1.5 mm or a local increase of the IMT of over 50% of the surrounding IMT value in any visualized segment of the carotid arteries was considered an ASP. The plaque score was calculated as follows: 0—no ASP in any segment of carotid arteries; 1—ASP with stenosis less than 30% of the vessel diameter; 2—ASP with stenosis at 30–50% of the vessel diameter or multiple ASPs with stenosis less than 30% of the vessel diameter; 3—ASP with stenosis over 50% of the vessel diameter or multiple ASPs with stenosis at 30–50% of the vessel diameter. The thickness of the intima–medial layer of the carotid arteries was measured in the distal segments of the common carotid arteries at three angles of interrogation, and the mean value of these measurements was used to characterize the carotid intima–medial thickness [41,42,43,44].

### 2.3. MtDNA Heteroplasmy Measurements

DNA from blood leukocyte samples was isolated using a phenol–chloroform method. PCR fragments of DNA containing the investigated regions of 10 mitochondrial genome mutations (m.12315G>A, m.5178C>A, m.652delG, m.14459G>A, m.3256C>T, 652insG, m.15059G>A, m.13513G>A, m.1555A>G, and m.14846G>A) were pyrosequenced. The heteroplasmy levels of these mtDNA mutations were analyzed using a quantitative method based on pyrosequencing technology developed by M. A. Sazonova and colleagues [14].

The size of the PCR fragments of DNA and primers for the PCR are listed in Table 1 [41,42].

In order to perform pyrosequencing of the PCR fragments of DNA, one primer from each of the primer pairs for the PCR was biotinylated. The total volume of qPCR reaction mixtures for each sample was 30 μL. The composition of the reaction mixture for qPCR: 0.4–0.6 μg mitochondrial DNA, 0.3 pM of each primer, 200 μM of each deoxyribonucleotriphosphate, 16.6 μM (NH_4_)_2_SO_4_, MgCl_2_ (1.5 mM for mutations m.14846G>A, m.15059G>A, and m.14459G>A; 2.5 mM for the rest of the investigated mutations), 67 mM Tris-HCl (pH 8.8), and 3 units of Taq polymerase. The qPCR was conducted using the PTC DNA Engine 200. In qPCR, the following annealing temperature was used for the primers:for mutations m.5178C>A, m.652delG, and m.652insG—60 °C;for mutations m.12315G>A, m.14459G>A, and m.1555A>G—50 °C;for mutations m.14846G>A, m.13513G>A, m.15059G>A, and m.3256C>T—55 °C.

The PCR fragments of DNA were analyzed on the automated pyrosequencing device PSQTMHS96MA (Biotage, Uppsala, Sweden) [41,45]. The primers for the pyrosequencing are listed in Table 2 [41,42].

To analyze the level of heteroplasmy mutations in the mitochondrial genome, we used the pyrosequencing method [19,20,21]. The given method is based on measuring the light intensity from the reaction of ATP and luciferin. This reaction is catalyzed by the luciferase enzyme. It should be noted that ATP is produced from pyrophosphate by sulfurase fermentation. The pyrophosphate, which gives its name to the entire pyrosequencing method, is in turn produced in a reaction between a nucleotide that is complimentary to a nucleotide in the assayed biotinylated one-chain fragment and the sequencing primer or the next nucleotide, which is also included in the complimentary chain. If the nucleotide added to the reaction medium is not complimentary to the assayed DNA fragment, there is no pyrophosphate production and, thus, no light induction. The light intensity is proportional to the quantity of the nucleotides built into the complimentary chain. For example, if one T nucleotide is present in a certain position in the chain, a peak corresponding to one portion of light will be seen on the pyrogram. And if there are two T nucleotides, there will be a double peak. It should be noted that pyrosequencing, unlike other sequencing methods, is designed for scanning a small DNA fragment (5 to 20 nucleotides) that contains the analyzed fragment and some control neighboring nucleotides. The sequence analysis begins from the place of connection of the DNA sequencing primer to the PCR fragment.

The quantitative assay of mtDNA mutations was conducted using peak-height analysis on the pyrogram in the studied region of the one-chain PCR fragment of the mitochondrial genome.

Analyzing differences in the peak sequence and size for homozygotes having 100% normal and 100% mutant alleles, we will be able to find the heteroplasmy level in the DNA sample for each particular mutation [19,20,21]. We used Formula (1) to calculate the heteroplasmy level:(1)$$P=\frac{h-N}{M-N}\cdot 100\%$$
where P is the heteroplasmy percentage;

h is the peak height for the studied nucleotide;

N is the peak height for the studied nucleotide corresponding to 100% of normal alleles in a sample; and

M is the peak height for the studied nucleotide corresponding to 100% of mutant alleles in a sample.

To calculate the total mutational burden, the quartile ranks of heteroplasmy level distributions were counted; the sum of the quartile numbers was used as an integral indicator of the total mutational burden. The quartile number for mutations that negatively correlated with carotid ASPs was taken with a negative sign.

### 2.4. Statistics

For statistical analysis of the obtained results, the software package SPSS 27.0 (IBM SPSS Statistics, IBM Corp., Endicott, NY, USA) was used [46]. Results were presented in terms of means and standard deviations. Correlation between the presence of carotid plaques and the level of mtDNA mutations was analyzed by Pearson’s correlation coefficient.

## 3. Results

In total, 452 participants were included in the study: 231 patients with atherosclerotic plaques in carotid arteries and 219 conditionally healthy study participants.

We created two samples of the study participants:

(1) Patients with atherosclerotic plaques in carotid arteries (the first sample);

(2) Conditionally healthy study participants (the second sample).

Patients with atherosclerotic plaques in carotid arteries were included in the first sample; conditionally healthy study participants were included in the second sample.

Clinical and laboratory characteristics of the study participants are presented in Table 3.

Table 3 demonstrates that the patients with atherosclerotic plaques in carotid arteries and the conditionally healthy study participants were comparable by age and gender. Patients with atherosclerotic plaques in carotid arteries had a significantly higher carotid intima–medial thickness than conditionally healthy study participants (0.828 (0.115) vs. 0.694 (0.100) mm, respectively; *p* < 0.001). For all other evaluated traditional cardiovascular risk factors, there was no significant difference between patients with atherosclerotic plaques in carotid arteries and the conditionally healthy study participants. All study participants had normal blood pressure, elevated body mass index (overweight, but not obese), borderline high levels of total cholesterol and low-density lipoproteins (LDLs), and low levels of high-density lipoproteins (HDLs). The mean carotid plaque score in the atherosclerotic group was 2.1 (0.7).

Next, the level of mtDNA heteroplasmy in blood samples of the study participants was evaluated, and the results are presented in Table 4. The heteroplasmy level (%) for the investigated mutations in each study participant is shown in the Appendix A (Appendix A) as a scatter plot for each of 10 investigated mtDNA mutation.

Table 4 shows that mean heteroplasmy levels of the mtDNA mutations m.5178C>A, m.652delG, m.12315G>A, and m.3256C>T were significantly higher in patients with atherosclerotic plaques in carotid arteries, while the mean heteroplasmy levels of the mtDNA mutations m.13513G>A, m.652insG, and m.14846G>A were significantly higher in conditionally healthy study participants. We analyzed the correlation of mean heteroplasmy level of mtDNA mutations with the presence of atherosclerotic plaques; the results are presented in Table 5.

According to Table 5, a significant positive correlation was revealed between atherosclerotic plaque size in carotid arteries and the heteroplasmy level of the mtDNA mutations m.652delG, m.12315G>A, m.3256C>T, and m.5178C>A, while the heteroplasmy level of the mtDNA mutations m.13513G>A, m.652insG, and m.14846G>A correlated negatively with the presence of carotid atherosclerotic plaques.

As the effect of different mutations is multidirectional, it is necessary to consider the cumulative impact of the 10 studied mutations or the total mutational burden. This characteristic was assessed in two stages:

(1) The construction of a logistic regression model (Table 6, Table 7 and Table 8);

(2) The construction of ROC curves (Table 9).

A predictor in the analysis of ROC curves was the probability of belonging to one category or another—0 (no atherosclerotic plaques) or 1 (the presence of atherosclerotic plaques of any size).

On the basis of values of the included variables, with the use of the model, the estimation of the probability of belonging to category “0” or “1” for each study participant was performed. The data obtained from the probability, which can be considered a measure of the relative risk, were used for the ROC analysis (Figure 1; Table 9).

According to the data from the ROC analysis, the model turned out to be significant. The evaluation of the predictive and explanatory power of the model for the total mutational burden in ASPs allowed us to consider that the predictive and explanatory power in the used model was significantly higher than that for the models constructed individually for each mutation. The total mutational burden of the 10 studied mitochondrial genome mutations was associated with 86.7% of ASPs. Thus, the complex evaluation of the 10 studied mutations of the mitochondrial genome made it possible to diagnose successfully at least 86.7% of carotid atherosclerosis cases. The sensitivity of this model was 0.754; the specificity was 0.862 (*p* ≤ 0.05). Specificity and sensitivity data for 10 investigated mutations are indicated in Appendix A. Specificity and sensitivity data for proatherogenic and antiatherogenic mutations in study participants from the Novosibirsk region are indicated in Appendix A.

In addition, the odds ratio was performed to evaluate the relative risk of ASP occurrence in the total sample. Table 10 shows that study participants from the Novosibirsk region with a combination of the mutations m.652delG, m.12315G>A, m.14459G>A, m.13513G>A, m. 3256C>T, m.1555A>G, m.14846G>A, m.652insG, m.15059G>A, and m.5178C>A have 4.5 times higher probability of carotid ASPs than in the total sample.

## 4. Discussion

In atherogenesis, blood cells play important roles. In the case of the occurrence of atherosclerosis, these blood cells migrate into the intima–medial layer through the endothelial layer. Monocytes play a signaling role in inflammatory and immune responses, and they then turn into macrophages. These macrophages eliminate the excess cholesterol, which accumulates in foci of atherosclerotic lesions.

One of the possible causes of atherosclerosis may be defects in the mitochondria of cells; due to this, there can occur a lack of adenosine triphosphate (ATP) in cells and tissues of the body, leading, according to the monoclonal hypothesis, to the unlimited proliferation of cells and the occurrence of atherosclerotic lesions [47,48,49,50,51].

Mitochondria of human cells are organelles that are surrounded by two membranes [52,53,54,55,56]. They contain their own genome, which is represented by circular double-stranded molecules comprising a combination of approximately 16,500 pairs nucleotides. Human mitochondrial DNA has 37 genes. Twenty-two of them encode transport RNA, 2 of them encode ribosomal RNA, and 13 of them encode protein subunits of the respiratory chain complex, to which belong cytochrome B, ATPase, cytochrome C oxidase, and NADH dehydrogenase [57,58,59,60,61].

Homo- and heteroplasmy are characteristic for mitochondria [62,63,64,65,66]. For homoplasmy, the presence of 100% normal copies of DNA in the mitochondria of a cell is characteristic. However, due to the instability of the mitochondrial genome, in the process of ontogenesis, mutations often appear in it. Perhaps it is connected with the fact that the process of breathing takes place specifically in mitochondria, which is why mitochondrial DNA can be damaged by reactive oxygen species (ROS). Some of the mutations can be of a hereditary nature, transferring with the mitochondria from a mother to her offspring. For heteroplasmy, the simultaneous presence of mutant and normal copies of the genome in mitochondria is characteristic. It is thought that somatic mitochondrial genome mutations can accumulate during the lifetime of an individual, leading to the occurrence and development of different diseases [41,42,45,62,67].

According to the literature, a link between some human pathologies with mitochondrial mutations has been detected. The heterogeneous group of system disorders, which are caused by defects in the functioning of mitochondria, are traditionally called mitochondrial cytopathies [68,69,70,71,72].

Such diseases, in most cases, cause damage to the muscular and nervous systems, and their clinical manifestations begin later. Defects in oxidative phosphorylation can occur in one or simultaneously in several protein complexes. In heteroplasmy, a cell can function normally for some period of time because there are normal copies of the mitochondrial genome. It is supposed that if the quantity of copies of mutant mitochondrial DNA is higher than a certain threshold value, the production of energy in mitochondria decreases and turns out to be significantly lower than the necessary level. In the case of an increase in the number of defective organelles, the need of a cell for energy grows, which causes the compensatory proliferation of mitochondria, including those that contain mutations [73,74,75,76,77].

This continues to worsen the situation, especially for those tissues in the organism that consume the largest amount of energy: muscular and nervous tissues. That is why an individual with heteroplasmy for a certain mutation may have a long asymptomatic period, which is characteristic for a range of mitochondrial cytopathies. If mutant copies of the mitochondrial genome accumulate in sufficient quantities, pathological characteristics begin to have manifestations. Though the manifestation of the pathology can begin at different ages, in the case of occurrence of the disease at an early age, the disease is more severe, and further predictions are less consoling [34,78,79,80,81].

Mitochondrial genome mutations causing human diseases are subdivided into structural transformations (large deletions, insertions, and duplications), single nucleotide substitutions, microdeletions, and microinsertions in genes of protein subunits of respiratory chain enzymes and transport and ribosomal RNA. Mutations such as single nucleotide substitutions, insertions, and deletions can be in coding and in non-coding regions of the mitochondrial genome. If these mutations are in an operator or in a promoter (i.e., a region of regulation of transcription), in a coding part of a mitochondrial gene, or if they can manifest themselves phenotypically, they can influence metabolic mechanisms [82,83,84,85,86].

With the aim of detecting the heteroplasmy level of mitochondrial genome mutations in the investigated samples, the author and colleagues worked out a new original method of quantitatively assessing mutant alleles of the mitochondrial genome based on the technology of pyrosequencing [41,45,82,87]. Using our new method, it turned out to be possible to detect the heteroplasmy level of both somatic and hereditary mtDNA mutations and also somatic mutations of the nuclear genome.

The method for the quantitative assessment of mutant alleles of the mitochondrial genome was worked out by M.A. Sazonova and her colleagues on the basis of pyrosequencing technology [41,45,82,87]. Pyrosequencing has a number of essential advantages compared with other quantitative methods including Sanger DNA sequencing, high-performance liquid chromatography, pyrosequencing, SnapShot, HRM (high-resolution melt profiling), temporal temperature-gradient gel electrophoresis (TTGE), the Invader assay, the amplification-refractory mutation system (ARMS), the endonuclease method using the Surveyor nuclease and next-generation sequencing on 454/Roche apparatus, the Applied Biosystems SOLiD apparatus, and a number of Illumina apparatuses applied to the analysis of mutations [88,89,90,91,92]. Pyrosequencing has the least number of errors and the highest number of advantages compared with the other methods of measurements of the percent of heteroplasmy of the mitochondrial genome. It represents the most unique opportunity for the analysis of a very short fragment of DNA containing the region of the investigated mutation. The DNA fragment size turned out to be, on average, 5–10 base pairs, which significantly lowers the probability of made mistakes during the analysis. As a result of this, the method developed by the Sazonova M.A. and her colleagues, based on pyrosequencing technology [41,45,82,87], could be a “gold standard” for all the other methods for detecting the percent of heteroplasmy in the mitochondrial genome, and it is advisable to use it for the verification of the level of heteroplasmy in mutations detected with the use of other methods.

The results of the present study revealed a significant difference in the level of mtDNA heteroplasmy between samples of patients with atherosclerotic plaques in carotid arteries and conditionally healthy study participants. The mitochondrial genome mutations m.5178C>A, m.652delG, m.12315G>A, and m.3256C>T were detected as proatherogenic since they were higher in patients with atherosclerotic plaques in carotid arteries, and the mtDNA mutations m.13513G>A, m.652insG and m.14846G>A were detected as antiatherogenic since they were higher in conditionally healthy study participants. These results were supported by a correlation analysis of the total group, which showed that the mutations m.652delG, m.12315G>A. m.3256C>T, and m.5178C>A had a highly significant positive correlation with the carotid plaque score, and the mutations m.13513G>A, m.652insG, and m.14846G>A had a highly significant negative correlation with the severity of carotid atherosclerosis.

It is noteworthy that the deletion of guanine at position 652 of the mitochondrial genome leads to a defect in the small subunit (12S) of ribosomal RNA. There is a high probability that the function of mutant ribosomes becomes suppressed or that they experience a complete dysfunction [41,45,82,87].

At the same time, the insertion of guanine at position 652 of the mitochondrial genome apparently stabilizes the 12S subunit of ribosomal RNA and, consequently, the ribosome. This can lead to the increased expression of protein chains of respiratory chain enzymes of the mitochondria, an increase in the number of these enzymes, and the launch of physiological processes in cells, which appear to protect the body from atherosclerosis [41].

Mutation m.3256C>T is localized in the 24th nucleotide of the tRNA-Leu gene (codon recognition UUR), which is the last nucleotide in the region of the termination of transcription. Termination happens in the case where a stop codon (UAG, UGA or UAA) appears at the A-site of the ribosome. If the tRNA corresponding to these codons is absent, the peptidyl–tRNA continues to be connected to the R-site of the ribosome. At this stage, the proteins RF1 or RF2 have the function of recognizing different codons and of catalyzing the disconnection of the polypeptide chain from the ribosome. Protein RF3 causes the dissociation of mRNA from the mitochondrial ribosome. When leucine is the last amino acid in a polypeptide, termination of transcription does not occur, and the polypeptide cannot disconnect from the mitochondrial ribosome. This leads to negative consequences such as the inability to use the subunits of ribosomes, which remain connected to polypeptides, for further synthesis. These polypeptides also cannot be used in metabolic processes. For this reason, the number of normal proteins in the respiratory chain in the mitochondria decreases, and the intensity of ATP synthesis falls [41,45,82,87].

In the single-nucleotide replacement mutant m.5178C>A, localized in the gene of the second subunit of NADH dehydrogenase, a leucine is replaced by a methionine. This mutation appears to lead to the dysfunction of the enzyme. This dysfunction contributes to the occurrence of atherosclerotic lesions. This is confirmed by the data from scientific literature [41,45,82,87,93].

The single nucleotide replacement in m.12315G>A affects nucleotide 52 of the transport RNA-Leu gene (codon recognition CUN), which is localized in the region of the stem of the T-loop. The universality of the formation of the tertiary structure of tRNAs is well known. During this process, loops T and D get closer and become connected to each other by the formation of additional links between the bases. Direct participation in such links involves semi-conservative and conservative residues. One such residue in transport RNA-Leu is the guanine at position 52 (functional group that connects to a purine ring; in this case, -O). In the case of the replacement of guanine by adenine (functional group that which connects to a purine ring; in this case, NH2), this causes disorder in the tertiary structure of the transport RNA, which leads to its dysfunction. As a result, complexes of transport RNA-Leu start to form less intensively, and, consequently, the possibility of the correct inclusion of leucine into polypeptide chains is reduced [41,45,82,87].

Mutation m.13513G>A leads to the replacement of asparagine acid by asparagine at position 393 of the fifth protein subunit of NADH dehydrogenase. The function of this region of the protein chain consists in the transmission of one electron in the respiratory complex to NADH on ubiquinone. In the case of the replacement of asparagine acid, which has an additional acid group having a negative charge, by asparagine, an amino acid in which the additional charge is absent because of two radicals that compensate for the charges of each other, a change in the level of the charge occurs in the region “Oxidored_q1”. This appears to lead to an increase in efficiency of the transmission of electrons to NADH on ubiquinone. Perhaps complex 1 of the respiratory chain starts to function more productively, and the production of energy increases. The authors of this article hypothesize that this mutation facilitates the improvement and stabilization of NADH dehydrogenase, which forms part of the mechanisms that protect cells from atherosclerotic lesions [41,45,82,87].

In mutation m.14846G>A, a glycine is replaced by a serine in cytochrome B, which is part of the third complex of the respiratory chain. Due to this replacement at the N-terminal end of cytochrome B, which is responsible for anchoring a protein in the membrane, an additional site is created for the phosphorylation of a protein chain. Apparently, a large number of mitochondrial genome copies containing this mutation inhibits the formation of atherosclerotic lesions and enhances the compatibility of mutant cells [41,45,82,87].

A search of published data enabled the discovery of a few studies devoted to revealing the mitochondrial genome mutations associated with increased risk of cardiovascular disease (CVD) development. In particular, association of the mitochondrial heteroplasmy variant 16223C>T with the increased presence of cardiovascular risk factors and premature myocardial infarction was demonstrated in a recent study [94]. Another study demonstrated the association of the mtDNA4977 deletion with an increased risk of long-term major adverse cardiac events and all-cause mortality in patients with stable coronary artery disease [95].

It should be noted that we conducted a comparison of the spectrum of mitochondrial genome mutations in the Moscow and the Novosibirsk regions. It turned out that for this parameter, these regions show some differences. In particular, in the Novosibirsk region, four mutations of mitochondrial genome associated with atherosclerotic plaques were detected (m.5178C>A, m.652delG, m.12315G>A, and m.3256C>T), while in the Moscow region, there were three proatherogenic mtDNA mutations (m.652delG, m.12315G>A, and m.14459G>A) detected. At the same time, in the Novosibirsk region, three mtDNA mutations associated with a protective effect in the case of the occurrence of atherosclerotic plaques (ASPs) (m.13513G>A, m.652insG, and m.14846G>A) were found, while in the Moscow region, only one (m.14846G>A) was found. At the same time, the mutations m.5178C>A, m.3256C>T, m.652insG, and m.13513G>A were also detected in the Moscow region but at a significance level of *p* ≤ 0.1, not of *p* ≤ 0.05. This may indicate the presence of a west–east gradient in the distribution of the mtDNA mutations m.5178C>A, m.3256C>T, m.652insG, and m.13513G>A.

Estimation of the total mutational burden from the 10 studied mutations of mitochondrial genome allowed the explanatory capacity of mitochondrial heteroplasmy for carotid atherosclerosis to be assessed.

Odds ratio analysis was performed to evaluate the relative risk of carotid atherosclerosis development in the total sample. Table 10 shows that study participants from the Novosibirsk region with the combination of mutations m.652delG, m.3336T>C, m.12315G>A, m.14459G>A, m.13513G>A, m. 3256C>T, m.1555A>G, m.14846G>A, m.652insG, m.15059G>A, and m.5178C>A have a 4.5 times higher probability of carotid ASPs than in the total sample.

The investigation of the total mutational burden is increasingly being used in clinical studies to assess the role of mitochondrial heteroplasmy in risk stratification for various pathologies. The combined estimation of the total mutational burden of the mitochondrial genome allowed the successful diagnosis of at least 86.7% of cases of ASP in carotid arteries from the Novosibirsk region. In our previous study, we determined that the total mutational burden due to mtDNA mutations explains 84.2% of atherosclerotic plaques in carotid arteries from the Moscow region [41]. The complex analysis of the results of previous studies allowed us to conclude that the number of heteroplasmic sites increases with age and has a negative correlation with the mtDNA copy number, a biomarker of mitochondrial dysfunction [96,97]. Several other studies aimed at comparing the total heteroplasmic burden in different age groups also found a strong association of the heteroplasmic burden with the age of participants and revealed a decreased mtDNA copy number in participants with a higher total heteroplasmic burden, which confirms the important role of mitochondrial heteroplasmy in the development of mitochondrial dysfunction [34,98,99]. It was observed in a study of over 6000 participants with different manifestations of CVD that the individual burden of heteroplasmic single-nucleotide variants increased with age and was most associated with the development of arterial hypertension [78].

The set of variants of mitochondrial heteroplasmy investigated in the current study (Novosibirsk region) was based on the previous findings (Moscow region) and allowed atherosclerosis-associated mtDNA mutations in the Novosibirsk region to be revealed [41].

When we compared the spectra of mitochondrial genome mutations in the Moscow and the Novosibirsk regions, a west–east gradient was found in the distribution of the mtDNA mutations m.5178C>A, m.3256C>T, m.652insG, and m.13513G>A.

## 5. Conclusions

In the analysis of samples from patients with atherosclerotic plaques in the carotid arteries and conditionally healthy study participants from the Novosibirsk region, four proatherogenic mutations of the mitochondrial genome (m.5178C>A, m.652delG, m.12315G>A, and m.3256C>T) and three antiatherogenic mutations of mtDNA (m.13513G>A, m.652insG, and m.14846G>A) were detected.

We detected a west-east gradient in the distribution of mtDNA mutations m.5178C>A, m.3256C>T, m.652insG and m.13513G>A during the comparison of the mitochondrial genome mutation spectra in the Moscow and the Novosibirsk regions.

The total mutational burden of the mitochondrial genome allowed the successful diagnosis of at least 86.7% of cases of atherosclerotic plaques in carotid arteries of patients from the Novosibirsk region, while in patients from the Moscow region, this parameter was 84.2%. Therefore, the total mtDNA mutational burden can be considered a promising marker for the timely prognosis of atherosclerosis development in patients from the Novosibirsk region.

## Figures and Tables

**Figure 1 biomedicines-12-01868-f001:**
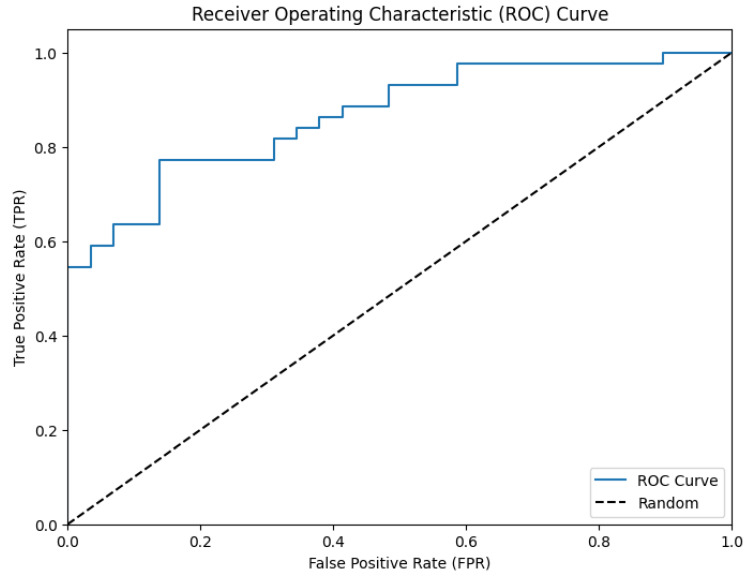
ROC curve of the interconnection of the mutational burden with atherosclerotic plaques in carotid arteries.

**Table 1 biomedicines-12-01868-t001:** The size of the PCR fragments of DNA and primers for the PCR.

Mutation	Primers	Size of DNA Amplicons
m.12315G>A	F: bio-CTCATGCCCCCATGTCTAA (12230–12249) R: TTACTTTTATTTGGAGTTGCAC (12337–12317)	108 bp
m.652delG	F: TAGACGGGCTCACATCAC (621–638) R: bio-GGGGTATCTAATCCCAGTTTGGGT (1087–1064)	467 bp
m.14459G>A	F: CAGCTTCCTACACTATTAAAGT (14303–14334) R: bio-GTTTTTTTAATTTATTTAGGGGG (14511–14489)	209 bp
m.5178C>A	F: bio-GCAGTTGAGGTGGATTAAAC (4963–4982) R: GGAGTAGATTAGGCGTAGGTAG (5366–5345)	383 bp
m.13513G>A	F: CCTCACAGGTTTCTACTCCAAA (13491–13512) R: bio-AAGTCCTAGGAAAGTGACAGCGAGG (13825–13806)	335 bp
m.652insG	F: TAGACGGGCTCACATCAC (621–638) R: bio-GGGGTATCTAATCCCAGTTTGGGT (1087–1064)	467 bp
m.3256C>T	F: bio-AGGACAAGAGAAATAAGGCC (3129–3149) R: ACGTTGGGGCCTTTGCGTAG (3422–3403)	294 bp
m.15059G>A	F: bio-CATTATTCTCGCACGGACT (14671–14689) R: GCTATAGTTGCAAGCAGGAG (15120–15100)	450 bp
m.1555A>G	F:TAGGTCAAGGTGTAGCCCATGAGGTGGCAA (1326–1355) R: bio-GTAAGGTGGAGTGGGTTTGGG (1704–1684)	379 bp
m.14846G>A	F: bio-CATTATTCTCGCACGGACT (14671–14689) R: GCTATAGTTGCAAGCAGGAG (15120–15100)	450 bp

**Table 2 biomedicines-12-01868-t002:** Primers for pyrosequencing.

Mutation	Primer
m.12315G>A	TTTGGAGTTGCAC (12328–12316)
m.652delG	CCCATAAACAAATA (639–651)
m.14459G>A	GATACTCCTCAATAGCCA (14439–14456)
m.5178C>A	ATTAAGGGTGTTAGTCATGT (5200–5181)
m.13513G>A	AGGTTTCTACTCCAA (13497–13511)
m.652insG	CCCATAAACAAATA (639–651)
m.3256C>T	AAGAAGAGGAATTGA (3300–3286)
m.15059G>A	TTTCTGAGTAGAGAAATGAT (15080–15061)
m.1555A>G	ACGCATTTATATAGAGGA (1537–1554)
m.14846G>A	GCGCCAAGGAGTGA (14861–14848)

**Table 3 biomedicines-12-01868-t003:** Clinical characteristics of study participants.

Characteristics	First Sample ^a^	Second Sample ^b^	Difference, *p*
Sex, male/female	61/170	70/151	0.182
Age, years	63.0 (5.9)	61.6 (5.0)	0.058
BMI, kg/m^2^	28.9 (4.9)	28.7 (4.8)	0.597
Diabetes, %	5	6	0.782
Systolic BP, mm Hg	130 (16)	130 (15)	0.900
Diastolic BP, mm Hg	80 (9)	81 (9)	0.817
Smoking, %	8	10	0.436
Carotid IMT, mm	0.828 (0.115)	0.694 (0.100)	<0.001 ^c^
Total cholesterol, mg/dL	229.5 (40.1)	234.2 (45.6)	0.248
HDL, mg/dL	49.1 (12.6)	50.3 (16.4)	0.387
LDL, mg/dL	157.6 (34.6)	160.6 (42.8)	0.414
TG, mg/dL	115.0 (60.7)	119.3 (80.6)	0.519

Note: Data presented as the mean (SD); ^a^ Patients with atherosclerotic plaques in carotid arteries; ^b^ Conditionally healthy study participants; ^c^ Statistically significant correlation at *p* ≤ 0.001.

**Table 4 biomedicines-12-01868-t004:** Mean level of mtDNA heteroplasmy (%).

mtDNA Mutation	First Sample ^a^	Second Sample ^b^	Difference, *p*
m.5178C>A	11.8 (9.6)	6.8 (7.8)	<0.001 ^c^
m.1555A>G	22.3 (13.1)	23.0 (10.3)	0.542
m.13513G>A	22.7 (11.9)	32.1 (16.2)	<0.001 ^c^
m.652delG	23.8 (11.0)	19.9 (10.2)	<0.001 ^c^
m.14846G>A	16.4 (15.7)	19.7 (16.6)	0.030 ^d^
m.12315G>A	35.5 (24.2)	31.1 (21.1)	<0.043 ^d^
m.14459G>A	5.1 (5.4)	4.2 (4.7)	0.055
m.652insG	1.4 (9.8)	3.9 (13.5)	0.027 ^d^
m.3256C>T	16.4 (16.5)	11.6 (8.6)	0.001 ^c^
m.15059G>A	11.2 (14.3)	9.8 (10.1)	0.226

Note: ^a^ Patients with atherosclerotic plaques in carotid arteries; ^b^ Conditionally healthy study participants; ^c^ Statistically significant correlation at *p* ≤ 0.001; ^d^ Statistically significant correlation at *p* ≤ 0.05.

**Table 5 biomedicines-12-01868-t005:** Correlation of mtDNA mutations with atherosclerotic plaques.

mtDNA Mutation	*r*, Pearson’s Correlation Coefficient	Significance, *p*
m.5178C>A	0.274	<0.001 **
m.1555A>G	−0.029	0.542
m.652delG	0.184	<0.001 **
m.12315G>A	0.097	0.041 *
m.14459G>A	0.091	0.055
m.3256C>T	0.180	<0.001 **
m.15059G>A	0.057	0.226
m.13513G>A	−0.316	<0.001 **
m.652insG	−0.105	0.027 *
m.14846G>A	−0.102	0.030 *

Note: *—statistically significant correlation at *p* ≤ 0.05; **—statistically significant correlation at *p* ≤ 0.001.

**Table 6 biomedicines-12-01868-t006:** Summary of a linear regression model of the interconnection of the mutational burden with atherosclerotic plaques in carotid arteries.

Model	Minus Twice the Log Likelihood	Cox and Snell R^2^	Nagelkerke R^2^
1	147,273	0.358	0.481 *

Note: *—The complex of features explains the dispersion of the dependent variable at 48.1%.

**Table 7 biomedicines-12-01868-t007:** Classification of cases of atherosclerotic plaque association with the total burden of the 10 mutations.

Model	Detected Cases		Predicted Cases		
			Association of Atherosclerotic Plaques with Total Burden of 10 Mutation		Percentage of Correct Predictions
			0.00	1.00	
1	Association of atherosclerotic plaques with total burden of 10 mutations	0.00	18	7	24.6
		1.00	11	37	50.6
	Total percentage value				75.3 *

Note: *—The percentage of correctly classified cases was 70%.

**Table 8 biomedicines-12-01868-t008:** Analysis of the included variables and the coefficient of the link force and direction.

Analyzed Variables							
	Mutations	B	S.E.	Wald	df	Sig.	Exp (B)
**Model 1**	m.1555A>G	−0.174	0.042	15.951	1	0.0001 **	0.840
	m.3256C>T	0.022	0.051	0.557	1	0.420	1.022
	m.14846G>A	−0.021	0.029	0.795	1	0.258	0.979
	m.5178C>A	0.034	0.045	0.760	1	0.355	1.034
	m.652delG	0.063	0.022	6.751	1	0.012 **	1.065
	m.12315G>A	0.133	0.017	21.961	1	0.0001 **	1.42
	m.13513G>A	−0.051	0.019	6.871	1	0.006 **	0.950
	m.14459G>A	0.031	0.015	2.862	1	0.043 **	1.031
	m.15059G>A	0.061	0.021	7.766	1	0.0001 **	1.062
	m.652insG	0.072	0.080	0.991	1	0.453	1.074
	Constant	−2.371	1.301	4.226	1	0.048 *	0.093

**Note:** (1) Coefficient B indicates the link direction; (2) **—Significant correlation of mutations with atherosclerotic plaques in carotid arteries (*p* ≤ 0.05); (3) *—Correlation of mutations with atherosclerotic plaques at a *p* ≤ 0.1 level of significance.

**Table 9 biomedicines-12-01868-t009:** ROC analysis of the interconnection of the mutational burden with atherosclerotic plaques in carotid arteries.

Probability of Faultless Prognosis				
Area under the Curve	Standard Error	Asymptomatic Significance	Asymptomatic Confidence Interval 95%	
			Lower than 95%	Higher than 95%
0.867	0.020	0.0015	0.771	0.910

**Table 10 biomedicines-12-01868-t010:** Odds ratio of cases of ASP in carotid arteries and their dependence on the total burden of mtDNA mutations.

OR	Risk of ASP in Carotid Arteries	Sensitivity	Specificity
4.5 (95% CI: 2.4–6.8)	2.08 (95% CI: 1.51–3.53)	0.749	0.858

## Data Availability

The datasets presented in this article are not readily available because the data are part of an ongoing study of patients from Russia with cardiovascular diseases.

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
