# Peer review of "Variability of Mitochondrial DNA Heteroplasmy: Association with Asymptomatic Carotid Atherosclerosis"

_biomedicines, 2024, doi:10.3390/biomedicines12081868_

Round 1

Reviewer 1 Report

Comments and Suggestions for Authors

The authors of the manuscript entitled "Variability of mitochondrial DNA heteroplasmy: association with asymptomatic carotid atherosclerosis" investigate the relations between heteroplasmy of ten mtDNA mutations and atherosclerotic plaques in asymptomatic patients.
The study highlights that 5 out of 10 variants analyzed are associated with the presence of plaques in patients, while 3 out of 10 variants are associated with a protective effect.

The study was carried out with a large number of participants (452) recruited from the Novosibirsk region (divided into two groups: 219 participants without carotid atherosclerosis and 231 with  carotid atherosclerotic plaques in the arteries), which makes the results statistically robust.
However, the manuscript has experimental limitations and further data must be provided to improve the work and make it suitable for publication.

First of all it is unclear how the 10 mtDNA mutations analyzed in this study were selected. Most likely they were chosen because they have been the subject of previous studies. This needs to be clearly stated in the text, with the appropriate references. If this is the case, a further 3 or 4 known mutations  not to be associated with CVD should be analyzed to make the study more valid.

The methods section should explain how the percentages of heteroplasmy of mtDNA mutations were determined.

For each mutation the percentage of heteroplasmy in each study participant should be shown (also in the supplementary data), for example as a scatter plot.

How were the ten mitochondrial genome mutations studied present in the patients? Are they single mutations or are they all present at the same time?

A table with specificity and sensitivity data for each mutation studied and for groups of proatherogenic and protective mutations should be reported (in the supplementary material, if appropriate).

Participants in the study were recruited from the Novosibirsk region, but in Section  4. Discussion, lines 239-251, data from Moscow patients are described. How did the authors obtain these data?

Are the mtDNA heteroplasmic levels in Table 4 given as percentages? If so, indicate this in the table.

The caption of the table should indicate the statistical test performed.

Author Response

Dear Reviewer 1, thank you for your perusal of the article and for recommendations and corrections made. Let me answer them point by point: Let me answer them point by point:

  1. We clarified in the article that the two study participant samples represent patients with atherosclerotic plaques in carotid arteries (the first sample) and conditionally healthy study participants (the second sample).
  2. According to your recommendation, we have inserted a fragment of text in the article: “10 mtDNA mutations analyzed in this study were selected because they have been the subject of previous studies in the Moscow region. The aim of the present study is the detection of regional mtDNA markers associated with atherosclerotic plaques (ASP) in carotid arteries in patients from the Novosibirsk region. The present study is replicative and its second aim is to detect the direction of the gradient of distribution of the analyzed mutations”.

In the Discussion section we have inserted a fragment of text: “It should be noted that we conducted a comparison of the spectrum of mitochondrial genome mutations in the Moscow and the Novosibirsk regions. It turned out that for this parameter these regions have some differences. In particular, in the Novosibirsk region, 4 mutations of mitochondrial genome associated with atherosclerotic plaques were detected (m.5178C>A, m.652delG, m.12315G>A and m.3256C>T), and in the Moscow region there were 3 proatherogenic mtDNA mutations (m.652delG, m.12315G>A, m.14459G>A) detected. At the same time, in the Novosibirsk region, three mtDNA mutations associated with a protective effect in case of occurrence of  atherosclerotic plaques (ASP) (m.13513G>A, m.652insG and m.14846G>A) were found, and in the Moscow region only one (m.14846G>A) was found.  At the same time, mutations m.5178C>A, m.3256C>T, m.652insG and m.13513G>A were also detected in the Moscow region, but at the significance level of p≤0.1, and not of p≤0.05. This may indicate the presence of a west-east gradient in the distribution of mtDNA mutations m.5178C>A, m.3256C>T, m.652insG and m.13513G>A.”

We plan to study additional mutations in our future work. In particular, we will analyze mtDNA mutations using the NGS method. It should be noted that the main direction of our angiopathology laboratory's research is the analysis of mutations in atherosclerosis and cardiovascular diseases. Therefore, we do not plan to study mutations in other pathologies at the moment.

  1. According to your recommendation, we have inserted in the methods section an explanation of how the percentages of heteroplasmy of mtDNA mutations were estimated: “The quantitative assay of mtDNA mutations was conducted using peak height analysis on the pyrogram in the studied domain of one chain PCR fragment of mitochondrial genome.

Analyzing differences in peak sequence and size for homozygotes having 100% normal and 100% mutant alleles, we will be able to find the heteroplasmy level in DNA sample for each particular mutation [19-21]. We used formula 1 to calculate heteroplasmy level:

                p=((h-N)/(M-N))X100% ,

where P is the heteroplasmia percent;

h is the peak height for the studied nucleotide;

N is the peak height for the studied nucleotide corresponding to 100% of normal alleles in a sample;

M is the peak height for the studied nucleotide corresponding to 100% of mutant alleles in a sample.”

  1. According to your recommendation, we have inserted a fragment of text in the article: “Heteroplasmy level (%) for the investigated mutations in each study participant shown in the supplementary materials (Figure supplementary 1) as a scatter plot for each mutation.”
  2. The tables with specificity and sensitivity data for each mutation studied and for groups of proatherogenic and protective mutations are presented in the supplementary material (Table supplementary 1 and Table supplementary 2).
  3. For the reason of instability of mitochondrial genome, in the process of ontogenesis, mutations often appear in it. Therefore, a patient may have all 10 mutations simultaneously. However, for the human body, those mutations that have reached or exceeded the threshold level of heteroplasmy (%) are important. The fact is that after this, the individual begins to develop atherosclerosis or a protective effect occurs in this disease.
  4. The heteroplasmy level of mtDNA mutations in Table 4 is given as percentages. We indicated this in the headline of the table.

  1. For statistical analysis of the obtained results the software package SPSS 27.0 (IBM SPSS Statistics, IBM Corp., NY, USA) was used. Results were presented in terms of means and standard deviations. Correlation between the presence of carotid plagues and level of mtDNA mutations was analyzed by Pearson’s correlation coefficient.
  2. We have highlighted all changes in the article in yellow.

Reviewer 2 Report

Comments and Suggestions for Authors

Overall, the research in the article is necessary, but there are still some issues that require further consideration. Can sensitivity and specificity be further improved? After all, for the current level of sensitivity and specificity, using the total mutational burden as a biomarker is still not very scientific.

1. The manuscript should be check carefully for typos errors, such as ‘oxigen’ in line 54. It is necessary to check and revise the entire article.

2. line 289: Is it appropriate to use low sensitivity and specificity as prognostic markers?

3. The concept of ‘total mutational burden’ should be clearly stated.

4. Is it appropriate to select 10 mitochondrial genome mutation sites? Do the 10 mitochondrial genome mutation sites have equal importance? Can sensitivity and specificity be improved by changing the number of mitochondrial genome mutation sites?

Comments on the Quality of English Language

Moderate editing of English language required

Author Response

Dear Reviewer 2, thank you for your perusal of the article and for recommendations and corrections made. Let me answer them point by point: Let me answer them point by point:

  1. We plan to expand the sample of study participants. We hope that the next study of the sample of study participants will have higher sensitivity and specificity rates. In the supplementary materials tables, we calculated sensitivity and specificity separately for each studied mutation (Table supplementary 1) and for groups of proatherogenic and antiatherogenic mutations (Table supplementary 2). We obtained quite high values, which indicate the feasibility of using the total mutation burden as a prognostic marker.
  2. We have corrected the text of the article “oxigen” for “oxygen”.
  3. In the supplementary materials tables, we calculated sensitivity and specificity separately for each studied mutation (Table supplementary 1) and for groups of proatherogenic and antiatherogenic mutations (Table supplementary 2). We obtained quite high values, which indicates the feasibility of using the total mutation burden as a prognostic marker.
  4. In our case, the total mutation burden is the combined effect of the threshold level of heteroplasmy of all 10 studied mutations on the occurrence of atherosclerotic plaques (in the case of proatherogenic mutations), or the occurrence of a protective effect in atherosclerosis (in case of antiatherogenic mutations).
  5. 10 mtDNA mutations analyzed in this study were selected because they have been the subject of previous studies in the Moscow region. The aim of the present study is the detection of regional mtDNA markers associated with atherosclerotic plaques (ASP) in carotid arteries in patients from the Novosibirsk region. The present study is replicative and its second aim is to find the direction of the gradient of distribution of the analyzed mutations.

When analyzing 10 mtDNA mutations in study participants from the Moscow region, we obtained high sensitivity and specificity rates [1].

  1. We have highlighted all changes in the article in yellow.

References

  1. Sazonova, M.A.; Sinyov, V.V.; Ryzhkova, A.I.; Galitsyna, E.V.; Khasanova, Z.B.; Postnov, A.Y.; Yarygina, E.I.; Orekhov, A.N.; Sobenin, I.A. Role of Mitochondrial Genome Mutations in Pathogenesis of Carotid Atherosclerosis. Oxid. Med. Cell. Longev. 2017;2017:6934394. doi: 10.1155/2017/693439437.

Round 2

Reviewer 1 Report

Comments and Suggestions for Authors

The authors have answered all my questions in detail and improved the manuscript. The manuscript can be accepted as it is now submitted.